# An Immuno-Separated Assay for Ochratoxin Detection Coupled with a Nano-Affinity Cleaning-Up for LC-Confirmation

**DOI:** 10.3390/foods11081155

**Published:** 2022-04-15

**Authors:** Jie-Biao Guo, Jin-Sheng Cheng, Tai-Long Wei, Fan-Min Wu, Gui-Hong Tang, Qing-Hua He

**Affiliations:** 1Provincial Key Laboratory for Utilization and Conservation of Food and Medicinal Research in Northern Guangdong, Shaoguan University, No. 288 Daxue Road, Shaoguan 512005, China; 2School of Innovation and Entrepreneurship, Shaoguan University, No. 288 Daxue Road, Shaoguan 512005, China; chengjs@sgu.edu.cn; 3State Key Laboratory of Food Science and Technology, Sino-Germany Joint Research Institute, Nanchang University, No. 235 Nanjing East Road, Nanchang 330047, China; ncuspyweitailong@163.com; 4Shaoguan Food and Drug Inspection Institute, No.13 Muxi Road, Shaoguan 512026, China; fm1226275663@163.com (F.-M.W.); tgh1221@163.com (G.-H.T.)

**Keywords:** immuno-separated assay, nano-affinity cleaning up, immuno-magnetic adsorbent, immuno-fluorescent reporters, LC-confirmation

## Abstract

An immuno-separated assay for ochratoxin A detection coupled with a nano-affinity cleaning up for LC-confirmation was developed. Firstly, ochratoxin A was modified to quantum dot beads for immuno-fluorescent reporters. Secondly, Fe_3_O_4_ magnetic nanoparticles were conjugated with protein G for immuno-magnetic adsorbents. The immuno-separation of fluorescent reporters by magnetic adsorbents could be completed by ochratoxin A, so the fluorescent reporters released from the immune complex indicate a linear correlation with the concentration of ochratoxin A. Furthermore, the immuno-separated ochratoxin A can be eluted from magnetic adsorbent for LC-conformation. The optimized assay showed results as follows: the quantitative range of the immuno-separated assay was 0.03–100 ng mL^−1^ of ochratoxin A. The recoveries for spiked samples ranged from 78.2% to 91.4%, with the relative standard deviation (RSD) being 11.9%~15.3%. Statistical analysis indicated no significant difference between the HPLC-FLD results based on commercial affinity column and by nano-affinity cleaning up.

## 1. Introduction

Ochratoxin A (OTA) is a worldwide mycotoxin that contaminates agricultural products and remains in processed foods such as cereals, spices, legume crops, dried fruits, fruit juices, coffee, beer, and baked goods. It is also can enter poultry and livestock products through feed, thus contaminating milk, meat, eggs, and their derivative foodstuffs [1].

OTA is considered to be seriously harmful to humans and animals due to its immunosuppressive, nephrotoxic, carcinogenic, and teratogenic effects [2]. According to its widespread distribution in the food chain and high stability during food processing, the maximum residual limits of OTA for various feeds and foods have been established by countries and regions all over the world (Table 1).

In order to monitor OTA for protecting the public from health risks, bioassays were used for large-scale screening OTA and chromatographic detections were consequently employed to confirm suspicious samples [3,4,5,6]. 

Bioassays for OTA screening improved in recent decades, becoming more convenient, sensitive, and accurate [7,8]. The most common bioassays for OTA screening are heterogeneous systems, as listed in Table 2. The immuno-chromatographic assays have the advantages of being rapid, sensitive, and cost-effective [9,10,11,12,13]. However, these membrane-based assays can only achieve semi-quantification [14,15]. Microplate-based assays and electrochemical biosensors have good sensitivity and quantitative accuracy but require cycles of washing and reagent adding that are not ideal for screening purposes [16,17,18,19,20,21,22,23].

Homogeneous bioassays for OTA are listed in Table 3. RET-based bioassays show high sensitivity and accuracy, with the convenience of non-washing, but suffer from the interference of the non-separation matrix [24,25,26,27,28,29,30,31]. Magnetic-separated bioassays have better sensitivity and reliability and benefit from OTA enrichment and matrix elimination [32,33,34,35,36,37].

When a sample containing OTA excess reaching a residual limit is screened out by bioassay, it is necessary to confirm the result of OTA by chromatographic detections [5]. Sample cleaning up is an important step before instrumental confirmation because it can protect instruments from damage and contamination and can help prevent signal interference from matrix effects [38,39]. Various reported schemes for cleaning up of OTA are compared in Table 4 [40,41,42,43,44,45,46,47,48]. It can be found that bio-recognitions for OTA cleaning-up show higher specificity and affinity than those for physical adsorptions [7]. Additionally, magnetic micro-cleaning up can provide faster kinetics and higher recovery of OTA purification than column-based sample adsorptions [49,50]. 

By learning from the reported experience, it was found that a magnetic-separated immunoassay can be coupled with micro-affinity cleaning up of OTA in a single process, as shown in the scheme of Figure 1. Based on this design, the immuno-separated assay on the homogeneous reaction of quantum reporters and magnetic adsorbents can improve the quantitative accuracy of OTA screening, and the following instrumental confirmation of its suspicious results could be more efficient.

## 2. Materials and Methods

### 2.1. Materials and Instruments

Recombinant Streptococcal Protein G, and Ochratoxin A, Aflatoxin B_1_, Zearalenone, Fumonisin B_1_, and Deoxynivalenol standards, were purchased from Sigma-Aldrich Chemical Co. (St. Louis, MO, USA). N-hydroxy-succinimide, N-(3-dimethylaminopropyl) -N′-ethylcarbodiimide, 2-(n-morpholine) ethanesulfonic acid (MES), ethanolamine, and gluconic acid were obtained from Aladdin Chemistry Co., Ltd. (Shanghai, China). Amino-modified CdSe/ZnS-based quantum dot beads (100 nm) with 630 nm emission wavelength were supplied by Kun Dao, Ltd. (Shanghai, China). Carboxyl-modified Fe_3_O_4_/SiO_2_ magnetic nanoparticle (200 nm) was purchased from Yi-Yan biotechnology Co., Ltd. (Luoyang, Henan, China). Antibody stabilizer solution was purchased from Tu-Feng, Ltd. (Shanghai, China). The immuno-affinity column of OTA was purchased from Casco Biotech Co. (China Hangzhou). The antibody of OTA was prepared in our laboratory. Other chemicals of analytical grade were purchased from Sinopharm Chemical Corp. (Shanghai, China).

Ultra-pure water was prepared using the Labonova LS10 reverse osmosis pure water system. (RODI, German). Confirmations of the immunoassays were carried out using an HPLC from Thermo Fisher Scientific Inc. (Waltham, MA, USA). Fluorescence spectra were obtained using a LUMINA fluorescence spectrophotometer from Thermo Fisher Scientific Inc. (Waltham, MA, USA). 

Assay buffer: 0.01 M PBS (pH 7.4) containing 10% methanol and 0.1% Triton X-100. Extraction buffer: 0.02 M PBS (pH 7.4) containing 60% methanol. Antibody working solution: a hundred-µL of ascites containing OTA monoclonal antibodies were diluted in 1.0 mL antibody stabilizer solution to obtain an antibody working solution.

Unhusked rice samples were collected at the local rice field harvest site. Beer samples were purchased at the local supermarket in Shaoguan city.

### 2.2. Methods

#### 2.2.1. Synthesis of Immuno-Magnetic Adsorbents (IMAs)

Immuno-magnetic adsorbents (IMAs) were prepared as in Figure 1b; 0.5 mg ethylcarbodiimide and 20 mg carboxyl-modified magnetic nanoparticles (carboxyl-MNPs) were mixed in 10 mL 0.05 M 2-(n-morpholine) ethanesulfonic acid (MES) buffer (pH 6.5) for 30 min with gentle shaking at 20 °C to activate the carboxyl group. Then, 10 µg of protein G was added to the system with another 30 min shaking for conjugation with activated carboxyl-MNPs to achieve synthesis. The excessive activated carboxyl groups of the IMAs were blocked using ethanolamine. The IMAs were purified by magnetic separation and being washed 5 times and were re-dispersed with 0.02 M PB (pH 7.4) and stored at 4 °C prior to use. 

#### 2.2.2. Preparation of Immuno-Fluorescent Reporter (IFR) 

Amino quantum dot beads (amino-QDBs) were coupled with OTA for synthesis of immuno-fluorescent reporters (IFRs) using an active ester protocol [30]. As in Figure 1c, freshly prepared ethylcarbodiimide (2.0 mg in 0.5 mL methanol) and N-hydroxy- succinimide (0.2 mg in 0.5 mL methanol) were mixed with OTA (20 µg in 200 µL methanol) for 120 min of shaking at 25 °C to obtain ester-activated OTA. To optimize the coupling ratio of OTA to IFRs, ester-activated OTA was mixed with 2.0 nmol (5.0 µg) amino-QDBs in 2.0 mL borate buffer (pH6.5) with gradient mole ratios of 1.5, 2.0 2.5, 3.0, and 4.0, respectively. After 4 h of gentle shaking at 25 °C, the excessive amino on the surface of IFRs was blocked with 0.1 mmol ester-activated gluconic acid. All IFRs were separated with centrifugation at 16,000× *g* for 30 min at 4 °C and then re-dispersed with 0.02 M phosphate buffer (PB, pH 7.4), stored at 4 °C prior to use. 

#### 2.2.3. Verification of Antibody Level for the Immuno-Separated Assay 

To verify impact of antibody level on the immuno-separated capacity of OTA, gradient levels of 3, 4, 5, 6, and 7 µL of antibody solutions and 3 µg of IMAs were added to 0.5 mL of assay buffer containing 20 ng of OTA for 10 min incubation at 25 °C.Then, IMAs were separated to elute OTA for HPLC-FLD detection. The antibody level that causes a desired immuno-separated recovery of OTA was the verified condition of the assay. 

#### 2.2.4. Optimization of Dosage of IFRs 

To optimize the dosage of IFRs, gradient dosages of 0.11, 0.12, 0.13, 14, and 0.15 µg of IFRs were added for immuno-separation under verified antibody level, respectively. The fluorescence remaining in supernatant was detected to verify whether the IFRs dosage was close to the immune equivalent of the verified antibody level.

#### 2.2.5. Quantitative Optimization of the Assay 

Verified IFRs dosage and antibody level were applied for the development of the assay. OTA standard solutions of 0, 0.03, 0.10, 0.30, 1.0, 3.0, 10, 30, and 100 ng mL^−1^ were applied for quantitative evaluation of the assay. The linear optimization was carried out by minor adjusting of the reporter level. 

#### 2.2.6. Process of the Immuno-Separated Assay

A 6-µL antibody solution and 3 µg of IMAs were mixed with 200 µL standard solution (or sample solution) and 800 µL assay buffer for 15 min incubation. Then, the IMAs were collected and re-dissolved with 350 µL assay buffer containing 0.13 µg of IFRs for 10 min incubation. After magnetic separation, 300 µL of supernatant was taken to detect fluorescence with 360 nm excitation and 630 nm emission. The concept of efficiency of competition (*E*) is applied for linear regression against OTA concentration. The value of E is defined as Formula (1): *E* = (*F*_*x*_ − *F*_0_)/*F*_0_
(1)


Herein, *F*_0_ is the fluorescence intensity of the remaining IFRs under separation by IMAs without the competition of OTA. *F_x_* is the fluorescence intensity of IFRs released from the immune complex under competition of OTA.

#### 2.2.7. Evaluation of Cross-Reactivity

Aflatoxin B_1_, Zearalenone, Fumonisin B_1,_ and Deoxynivalenol at concentrations of 10, 100, and 1000 ng mL^−1^ were detected by the assay, respectively. The cross-reactivity of each mycotoxin was evaluated by analyzing the effects of concentration on the immuno- separation.

#### 2.2.8. Detection of Spiked Samples

Rice samples, being confirmed as not containing OTA by HPLC-FLD, were spiked with OTA to levels of 5, 10, and 20 ng g^−1^, respectively, and 5.0 g of OTA-spiked rice sample was extracted with 5.0 mL of extraction buffer by vortex shaking for 15 min. The sample was centrifuged at 9000× *g* for 10 min, and 200 µL of supernatant was taken for the immuno-separated assay according to the process of 2.2.6. OTA-spiked beer samples were diluted with extraction buffer and detected the same as above. 

#### 2.2.9. Nano-Affinity Cleaning up for LC-Confirmation

To confirm a suspicious result of the immuno-separated assay, the collected IMAs were eluted by 200 µL 80% acetonitrile solution (containing 2% acetic acid) for twice. The two eluting solutions were combined and filtered using a 0.1µm membrane for HPLC-FLD detection, according to reference [46]. 

## 3. Results 

### 3.1. Evaluation of Nonspecific/Specific Binding between IFRs and IMAs 

As shown in Figure 2a, the magnetic separation percentage of fluorescent reporters (IFRs) by magnetic adsorbents (IMAs) without the OTA antibody was less than 3%, which indicated that the nonspecific binding between IFRs and IMAs was acceptable. Specific binding evaluation showed that the separation percentage of IFRs by IMAs with the OTA antibody was greater than 93%, indicating that the specific separation of IFRs by IMAs met all expectations. The TEM of magnetic separated immune complex in Figure 2b identified successful immuno-binding between IFRs (100 nm) and IMAs (200 nm).

### 3.2. Optimization of OTA Coupling Ratio in IFRs 

As Figure 3a, with the increase of the molar ratio of OTA/amino-QDBs for reaction, the immuno-separated percentage of IFRs by IMAs improved. When the molar ratio of OTA/amino-QDBs was higher than 2.5, the immuno-separated percentage reached an immune equilibrium. As can be seen in Figure 3b, TEM showed that each IFR (100 nm) bound one IMA (200 nm), which proved the IFRs were in univalent modification under OTA/amino-QDBs molar ratio of 2.5 for reaction. As seen in Figure 3c, when the mole ratio OTA/QDBs for reaction was higher than 2.5, IFRs were coupled with multivalent OTA that could result in the aggregation of IFRs and IMAs. The aggregation between multivalent modified IFRs and IMAs could make the competition of immuno-separation by OTA insensitive. Thus, the OTA/QDBs ratio of 2.5 was the optimal condition for IFRs preparation.

### 3.3. Kinetics Analysis

As shown in Appendix A, the immuno-separated kinetics of OTA and IFRs by the IMAs reached an immune balance after 10 min. Thus, the magnetic immuno-separated process of OTA and IFRs was identified to be 10 min. 

### 3.4. Optimization of the Immuno-Separated Assay

#### 3.4.1. Impact of Antibody Level on Detection Capacity

The impact of antibody level on the separated capacity of OTA was evaluated by HPLC-FLD following a nano-affinity separation. As seen in Figure 4a, the application of 6 µL antibody solution caused 96.5% of the separated recovery of 20 ng OTA that reached immune equilibrium because further increasing the antibody level had little effect on the recovery. Thus, 6 µL of antibody solution could provide the adequate separated capacity of OTA in the immuno-separated assay.

#### 3.4.2. Effect of IFRs Dosage on Quantitative Performance

As seen in Figure 4b, the appropriate remaining percentages were obtained after the immuno- separation of 0.13 µg and 0.14 µg of IFRs by 6 µL of antibody solution. Accurately, 0.13 µg and 0.14 µg of IFRs were used to develop immuno-separated assays in coordination with 6.0 µL of antibody solution, respectively. As seen in Figure 5a,b, two immuno-separated assays indicated the same quantitative range of 0.03–100 ng mL^−1^ with good linear correlation. Comparing the slopes of the two linear equations, the sensitivity of the assay using 0.13 µg IFRs is higher than the other. Therefore, 0.13 µg of IFRs was verified as being the optimal dosage for the immuno-separated assay.

#### 3.4.3. Evaluation of Cross-Activity

In terms of Appendix A, the increasing of mycotoxin concentrations had no effect on the immuno-separation in the assay. Thus, no obvious cross-reactivity against Aflatoxin B_1_, Zearalenone, Fumonisin B_1,_ and Deoxynivalenol of the assay was obtained.

### 3.5. Immuno-Separated Assay of Spiked Samples

As shown in Figure 6a, the immuno-separated assay obtained recoveries of 82.4%–87.5%, with RSD of 11.9%–15.3% for rice samples in three OTA spiked levels. As shown in Figure 6b, the immuno-separated assay showed recoveries of 78.2%–91.4%, with RSD of 13.3%–14.7% for beer samples in three OTA spiked levels. The precision and accuracy of the immuno-separated assay met expectations.

### 3.6. Nano-Affinity Cleaning up for LC-Confirmation

The nano-affinity cleaning up coupled with the immuno-separated assay was carried out for LC-FLD confirmation of rice and beer samples, which was described in Appendix A. At the same time, a commercial immuno-affinity column (IAC) cleaning up was applied for parallel comparison, which was described in Appendix A. As described in Table 5, the LC-FLD coupled with the nano-affinity cleaning up obtained recoveries of 88.4%–90.1% with RSD of 0.92%–10.7%. Additionally, the LC-FLD coupled with affinity column cleaning up obtained recoveries of 80.6%–98.5%, with RSD of 0.98%–6.81%. The results showed no significant difference between the pretreatment by IAC and by nano-affinity cleaning up. Both of the LC-FLD confirmations exhibited good agreement with the current immuno-separated assay. 

## 4. Discussion

### 4.1. Nonspecific/Specific Binding between IFRs and IMAs 

Because amino-QDBs and carboxyl-MNPs were applied for synthesis of IFRs and IMAs, it is important to eliminate the nonspecific binding of the positive/negative charge between them. So, gluconic acid was used to block the amino group of IFRs, and ethanolamine was used to block carboxyl groups of IMAs, respectively. As shown in Figure 2c, Zeta potential detections showed that the positive charge of IFRs (+0.09 mV) was obviously lower than amino-QDBs (+0.35 mV) and the negative charge of IMAs (−0.15 mV) was obviously lower than carboxyl-MNPs (−0.42 mV). The nonspecific binding of IFRs and IMAts was successfully controlled to a low level, which proved that the blocking of amino and carboxyl groups by chemical modification resulted in charge elimination. 

### 4.2. Optimization of OTA Coupling Ratio in IFRs 

To optimize the coupling ratio of OTA in the IFRs, the gradient molar ratios of OTA/QDBs were used for the synthesis of IFRs. When the reaction molar ratio was lower than 2.5, the immuno-separation of IFRs showed that some of them were not modified with OTA. Under the reaction molar ratio of higher than 2.5, TEM results indicated crosslinked immune complex between multivalent modified IFRs and IMAs. The irregular result of the immuno-separated assay due to immune aggregation by excessive modification of IFRs was the confusion in the early study. 

The immuno-separated percentage of IFRs and TEM results showed that IFRs were modified with univalent OTA under the reaction ratio of 2.5. The regular immune complex released the IFRs from the immuno-separation in a good quantitative response to the competition of OTA.

### 4.3. Impact of Antibody Level on Detection Capacity 

The immuno-separated capacity of the assay relied on the amount of antibodies that provide immune binding sites. Verified results indicated that 6 µL antibody solution was the immune equivalent of 20 ng of OTA. According to the scheme of the current system, this verified antibody level could result in a detection capacity of 20 ng of OTA in the immuno-separated assay and the following nano-affinity cleaning up.

### 4.4. Effect of Reporters Dosage on Quantitative Performance

Because the (*F*_*x*_ − *F*_0_)/*F*_0_ value was employed to reflect the competition effect of OTA to the immuno-separation, the optimization of the IFRs dosage to obtain an appropriate *F*_0_ value is important for the quantitative performance of the assay. Appropriate *F*_0_ values were obtained using 0.13 µg and 0.14 µg of IFRs for immuno-separation by 6 µL of antibody solution. Immuno-separated assays based on these two *F*_0_ values showed a satisfied linear correlation of (*F*_*x*_ − *F*_0_)/*F*_0_ in response to the competition of OTA. The dosages of 0.11 µg and 0.12 µg of IFRs resulted in overly low *F*_0_ values in the same immuno-separated condition that could cause an irregular (*F*_*x*_ − *F*_0_)/*F*_0_ value in response to the competition of OTA. As for the application of 0.15 µg of IFRs, a too high *F*_0_ value made the response of the (*F*_*x*_ − *F*_0_)/*F*_0_ value to competition of OTA insensitive. 

Results indicated that a minor excess dosage of IFRs to the antibody level could obtain good linear correlation of (*F*_*x*_ − *F*_0_)/*F*_0_ value to the OTA concentration. The (*F*_*x*_ − *F*_0_)/*F*_0_ value can reflect the competition effect of OTA on immuno-separation. 

### 4.5. Immuno-Separated Assay of Spiked Samples 

The worldwide MRLs of OTA for most commodities range from 5 to 20 ng mL^−1^. The spiking levels of 5–20 ng mL^−1^ were appropriate for the evaluation of the performance of the assay. As described in Figure 6a,b, the linear equations of immuno-separated assays in different detections were slightly different from each other. Thus, the data of the immuno-separated assay for standards and samples are comparable only in parallel detection. The results of the immuno-separated assay in the detection of spiked samples resulted in precision and accuracy benefits for the homogeneous immune reaction between nano-particles of IFRs and IMAs in liquid phase.

### 4.6. Nano-Affinity Cleaning up for LC-Confirmation

The results showed no significant difference between the LC-FLD confirmations by commercial immune affinity column and by nano-affinity cleaning up. The magnetic nano-affinity cleaning up had better rapidity and was simpler than the affinity column. 

The separated capacity of this nano-affinity is 20 ng of OTA, which is equal to that of its coupling immuno-separated assay. Compared to the affinity capacity of IAC being 100 ng of OTA, this nano-affinity cleaning up requires a lower number of antibodies so that it is more cost-effective. By being eluted with a total of 400 µL of buffer, the detection capacity of LC confirmation is 50 ng mL^−1,^ which can meet the requirements of OTA residual legislations.

## 5. Conclusions

An immuno-separated assay for OTA detection, coupled with a nano-affinity cleaning up for LC-confirmation, was developed based on the following experiences. First, modification of the amino of IFRs and the carboxyl group of IMAs by small molecular reagents helped to control nonspecific adsorption. Second, the optimization of univalent coupling IFRs made the immuno-separation could be sensitively competed by OTA. Third, the effect of the OTA antibody level on the detection capacity was verified by the immuno-separation result. Fourth, the minor excess dosage of IFRs compared to the antibody level could improve the quantitative performance of an immuno-separated assay.

The optimized immuno-separated assay was based on 6.0 µL of antibody solution and 0.13 µg of IFRs and had a quantitative range of 0.03–100 ng mL^−1^ with a linear correlation of 0.9872. The recoveries in the detection of rice and beer samples in OTA spiking levels of 5.0, 10, and 20 ng mL^−1^ by the immuno-separated assay ranged from 78.2% to 91.4%, with RSD of 11.9%~15.3%. The LC-FLD confirmations of the OTA spiked samples by nano-affinity cleaning up coupled with the immuno-separated assay obtained a recovery range of 88.4%~90.1% with RSD of 0.92%~10.7%, which showed good agreement with the cleaning up by commercial IAC.

## Figures and Tables

**Figure 1 foods-11-01155-f001:**
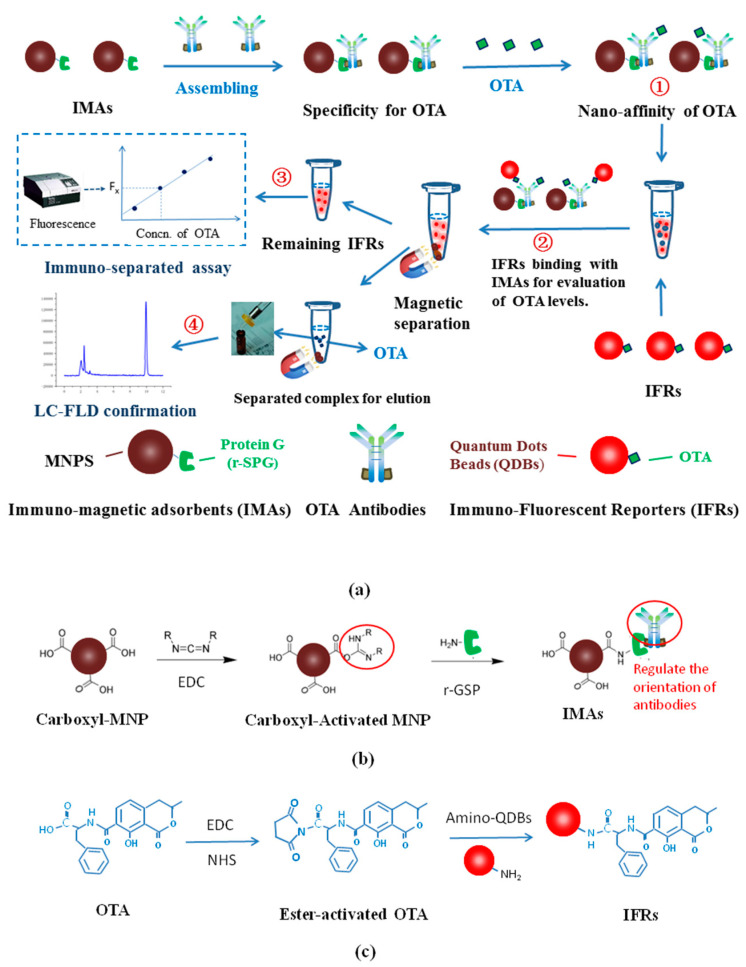
The scheme of the current experiment. (**a**) The scheme of the immuno-separated assay coupled with nano-affinity cleaning up for LC-confirmation. (**b**) The scheme of IMAs synthesis. (**c**) The scheme of IFRs synthesis. Notes of Figure 1a: ① Immuno-separation of OTA; ② Immuno-separation of IFRs, ③ Detection of fluorescence of remaining IFRs for Immuno-separated assay; ④ Elution of separated IMAs for LC-confirmation of OTA concentration.

**Figure 2 foods-11-01155-f002:**
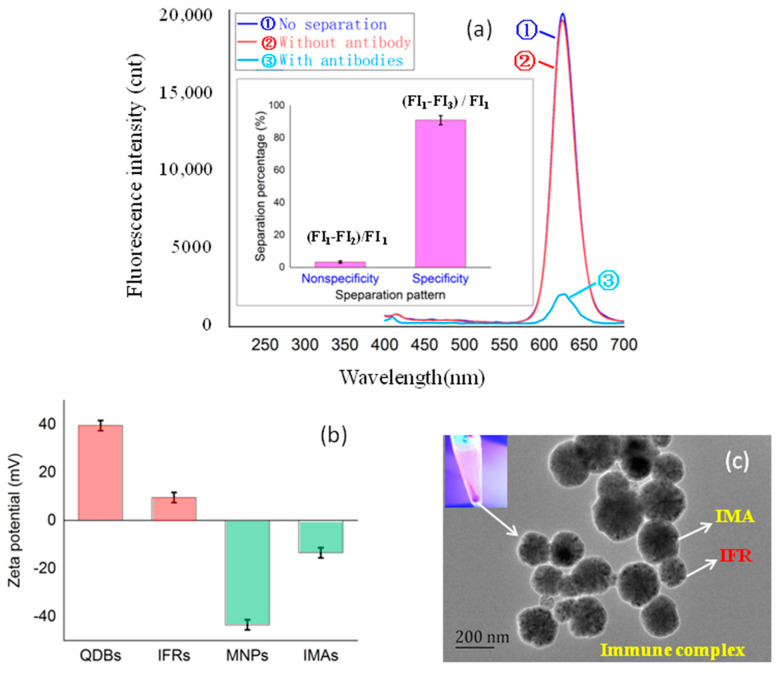
The evaluation of the performance of IFRs and IMAs. (**a**) The evaluation of specific/nonspecific binding between IFRs and IMAs. (**b**) The Zeta potentials of IFRs and IMAs. (**c**) The TEM of separated immune complex of IFRs (100 nm) and IMAs (200 nm).

**Figure 3 foods-11-01155-f003:**
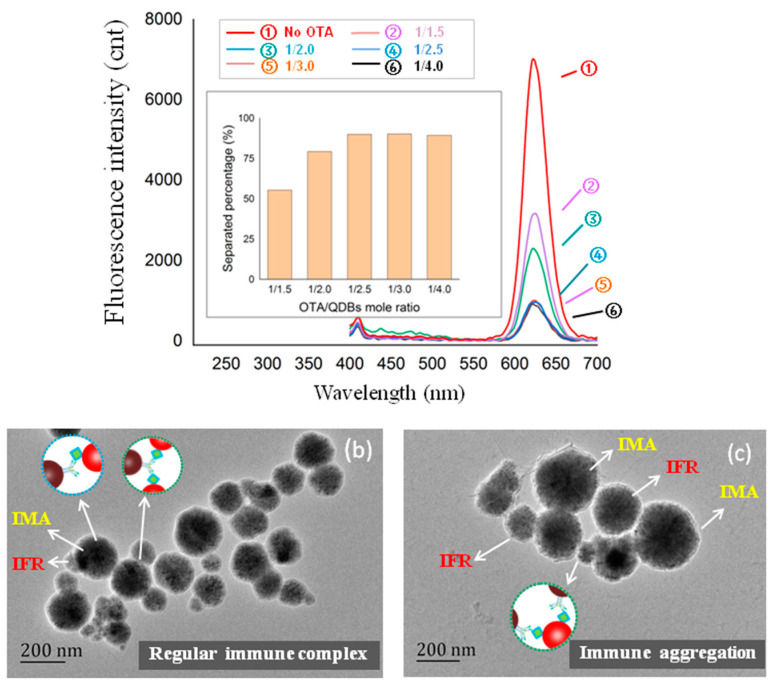
The optimization of the OTA coupling ratio in IFRs. (**a**) The effect of reaction molar ratio of OTA/QDBs on the immuno-separation ratio of IFRs by IMAs. (**b**) The TEM of a regular complex of univalent IFRs (100 nm) and IMAs (200 nm). (**c**) The TEM of the immune aggregation of multivalent IFRs (100 nm) and IMAs (200 nm).

**Figure 4 foods-11-01155-f004:**
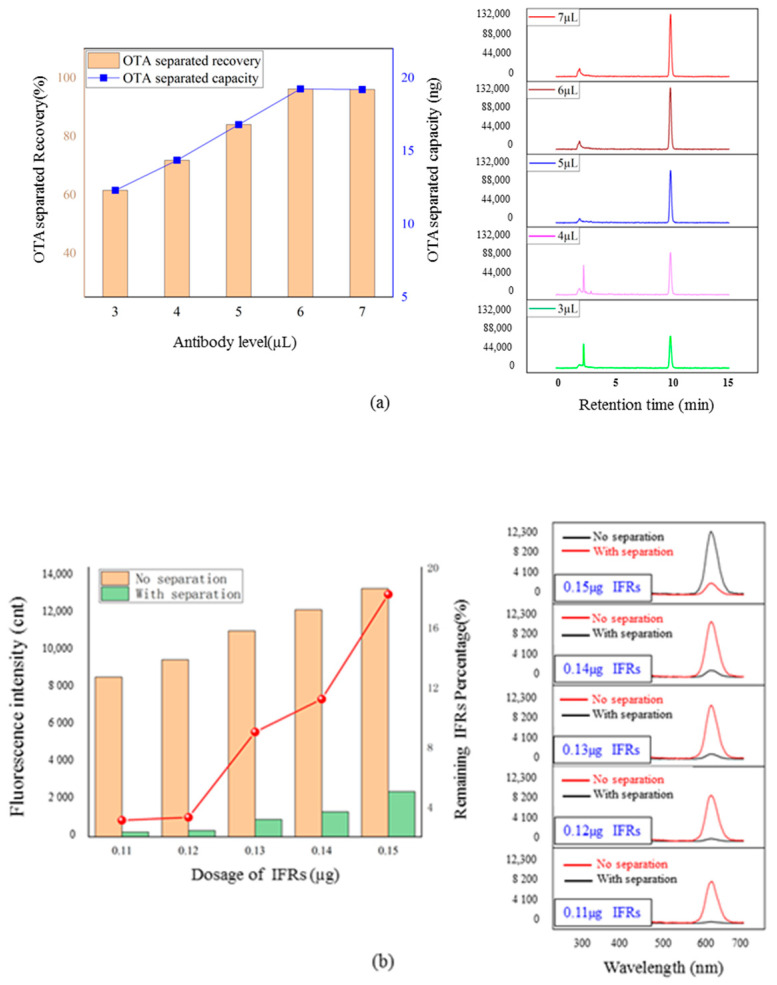
The optimization of IFRs dosage and antibody level for the immuno-separated assay. (**a**) Effects of antibody level on separated recovery for 20 ng of OTA. (**b**)Separation effects of different dosages of IFRs by 6 µL of antibody solution.

**Figure 5 foods-11-01155-f005:**
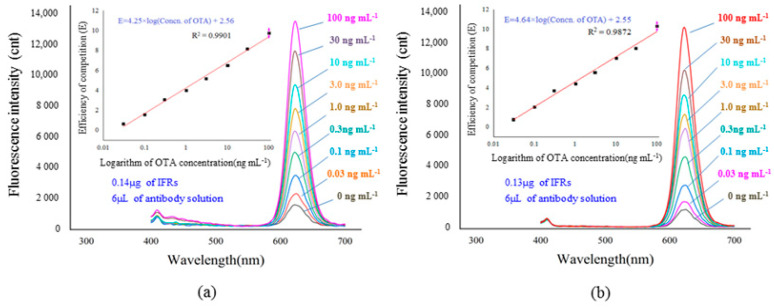
The standard curves of immuno-separated assays by different conditions. (**a**) The system based on 0.14 µg of reporters and 6 µL of antibody solution. (**b**) The system based on 0.13 µg of reporters and 6 µL of antibody solution.

**Figure 6 foods-11-01155-f006:**
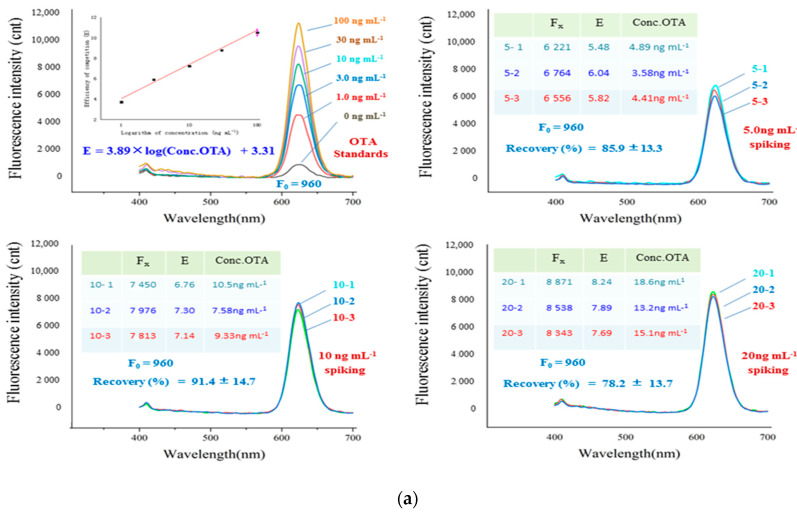
The detection of spiked samples by immuno-separated assay. (**a**) The detection of rice samples spiked with OTA. (**b**) The detection of beer samples spiked with OTA.

**Table 1 foods-11-01155-t001:** An overview of the worldwide legislation on OTA.

Commission and Country	Commodities	Maximum Residual Limits (µg kg^−1^)
Codex Alimentarius Commission	Cereals, processed cereals, dried fruits, coffee, cocoa, grape juice, wine, beer, spices, and bakery products.	5.0
European Commission		0.5–10 ^a^
Brazil		2.0–30 ^a^
China, Russia		5.0
India		20

^a^ Depends on different commodities (lowest–highest MRL).

**Table 2 foods-11-01155-t002:** Representative heterogeneous bioassays for rapid screening of Ochratoxin A.

Source	Detection Mechanism	LOD Value (µg kg^−1^)	Linear Range (µg kg^−1^)
Laura et al. (2011)	ICA using gold nanoparticles for labeling ^a^	1.5	Semi-quantitation
Majdinasab et al. (2015a)	Gold nanoparticle-based ICA ^a^	0.2	Semi-quantitation
Hao et al. (2021)	Magneto-gold nanohybrid-enhanced ICA ^a^.	0.094	Semi-quantitation
Majdinasab et al. (2015b)	Time-resolved fluorescent ICA^a^	1.0	Semi-quantitation
Majdinasab et al. (2019)	ICA by europium nanoparticle for labeling ^a^	4.0 × 10^−4^	Semi-quantitation
Zhou et al. (2021)	ICA by CdSe/ZnS QDs for labeling ^a^	0.07	Semi-quantitation
Perrotta et al. (2012)	EC immunosensor by voltammetry detection ^b^	0.008	0.01–20
Hao et al. (2020)	Photo-electrochromic visualization biosensor ^b^	0.290	1.0–500
Zhu et al. (2020)	EC sensor by labelled aptamer for signaling ^b^	0.0033	0.01–10.0
Pei et al. (2018)	ELISA on urease-induced gold nanoflowers ^c^	0.040	0.005–0.64
Sun et al. (2019)	Biotin/streptavidin nanobody-based ELISA ^c^	0.138	0.034–0.46
Mukherjee et al. (2021)	Chemiluminescence (CL) aptamer-ELISA ^c^	0.84 × 10^−3^	10^−3^–10^3^
Chen et al. (2021))	CuS based Chemiluminescence (CL) ELISA^c^	0.01	0.1–100

^a^ Immunochromatographic assays; ^b^ electrochemical immunoassay; and ^c^ micro-plate based ELISA.

**Table 3 foods-11-01155-t003:** The recent homogeneous bioassays for rapid screening of Ochratoxin A.

Source	Detection Mechanism	LOD Value (µg kg^−1^)	Linear Range (µg kg^−1^)
Dai et al. (2017)	Aptasensor by RET from UCNPs to graphene ^a^	0.001	0.001–250
Tang et al. (2019)	Nanobody-based RET immunoassay ^a^	0.06	0.1–10
Tian et al. (2020)	Nanoceria/graphene QDs RET nanosensor ^a^	2.5 × 10^−3^	0.01–20
Bi et al. (2020)	Aptasensor by RET on graphitic QDs/CoOOH ^a^	0.5 nM	1–140 nM
Kim et al. (2020)	Aptasensor by RET on UCNPs/Gold nanocap ^a^	0.022	0.1–1000
Zhang et al. (2013)	Magnetic aptasensor based on Tb^3+^ fluorescent ^b^	0.020	Not mentioned
Dai et al. (2016)	Magnetic aptasensor on upconversion fluorescent ^b^	0.005	0.01–100
Yan et al. (2020)	Magnetic aptasensor on catalyzing luminol ^b^	0.041	Not mentioned

^a^ RET-based homogeneous bioassay; ^b^ magnetic homogeneous bioassay.

**Table 4 foods-11-01155-t004:** The recent cleaning-up of Ochratoxin A for instrumental detections.

Source	Mechanism of Cleaning-up	Separation and Determination	LOD (µg kg^−1^)/LOQ (µg kg^−1^)	Recovery (%)
Cao et al. (2013)	Molecularly imprint (MIP)-based solid phase cleaning-up	UPLC-FLD	0.09/0.30	87.6–94.5
Duarte et al. (2013)	Immunoaffinity column (IAC)-based cleaning-up	LC–ESI-MS_2_	0.06/0.19	98.5–100.6
Ye et al. (2019)	Immunoaffinity magnetic beads coupled to UPLC-FLD	UPLC-FLD	0.24/0.80	86.3–95.4
Zhu et al. (2016)	Reversed phase/strong anion-exchange mixed-mode column	HPLC-FLD	0.006/0.02	81.6–100.8
Mashhadizadeh et al. (2013)	Fe_3_O_4_ nanoparticles coated with functional group for MSPE ^a^	HPLC-FLD	0.03/0.11	87–93
Turan & Şahin (2016)	Molecularly imprinted biocompatible magnetic nanoparticles	UV spectrophotometer	0.374/1.247	97.1–97.4
Armutcu et al. (2018)	P(HEMAPA)-4 monolithic column cleaning up	On-line 2D-HPLC	0.021/0.064	104.34–107.33
Chen et al. (2018)	Aptamer and affinity monolith dual selective extraction	HPLC-FLD	0.025/0.045	Higher than ‘sol-gel’ SPE.
Campone et al. (2018)	Automated on-line SPE by Oasis MAX column	HPLC–MS/MS	Being compliant with EU regulation N.519/2014	
Chen et al. (2019)	Hydrophilic aptamer-based hybrid affinity monolith	HILC ^b^	Not mentioned	94.9–99.8
Luci. (2020)	Molecularly imprinted solid phase column (MISPE)	HPLC-FLD	0.001/0.003	>89
Lyu et al. (2020)	Aptamer/MIP monolithic double-recognized column	HPLC-FLD	0.07/not mentioned	95.5–105.9

^a^ Magnetic solid phase extraction. ^b^ Hydrophilic interaction liquid chromatography.

**Table 5 foods-11-01155-t005:** The recoveries of OTA spiked samples detection by immuno-separated assay compared to LC-FLD with nano-affinity and IAC cleaning up.

	Immuno-Separated Assay (*n* = 3)	HPLC-FLD with Nano-Affinity Cleaning up (*n* = 3)	HPLC-FLD with IAC Cleaning Up (*n* = 3)
Detection of 5 ng g^−1^ spiked sample (ng g^−1^) ^a^	3.71, 3.93, 4.83	4.47, 4.22, 4.42	4.10, 4.19, 4.18
Mean recovery ± RSD (%) ^a^	83.1 ± 11.9	87.4 ± 2.64	83.2 ± 0.98
Detection of 10 ng g^−1^ spiked sample (ng g^−1^) ^a^	7.14, 9.62, 7.96	9.15, 8.42, 8.43	10.01, 9.71, 9.78
Mean recovery ± RSD (%) ^a^	82.4 ± 12.6	86.7 ± 4.18	98.3 ± 1.56
Detection of 20 ng g^−1^ spiked sample (ng g^−1^) ^a^	18.6, 14.5, 20.4	18.10, 17.81, 18.15	20.41, 19.42, 19.28
Mean recovery ± RSD (%) ^a^	87.5 ± 15.3	90.1 ± 0.92	98.5 ± 3.08
Detection of 5 ng g^−1^ spiked sample (ng g^−1^) ^b^	3.58, 4.98, 4.41	4.50, 3.61, 4.58	4.09, 4.03, 3.96
Mean recovery ± RSD (%) ^b^	85.9 ± 13.3	84.6 ± 10.7	80.6 ± 1.41
Detection of 10 ng g^−1^ spiked sample (ng g^−1^) ^b^	7.58, 10.5, 9.33	9.12, 8.57, 8.53	10.53, 9.29, 9.42
Mean recovery ± RSD (%) ^b^	91.4 ± 14.7	87.4 ± 3.29	97.5 ± 6.81
Detection of 20 ng g^−1^ spiked sample (ng g^−1^) ^b^	18.6, 15.1, 13.2	17.64, 17.94, 16.70	19.63, 18.89, 18.06
Mean recovery ± RSD (%) ^b^	78.2 ± 13.7	87.2 ± 3.24	94.4 ± 3.92

^a^ Spiked rice samples; ^b^ spiked beer samples; and Relative Standard Deviation (RSD).

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
