# Peer review of "An Immuno-Separated Assay for Ochratoxin Detection Coupled with a Nano-Affinity Cleaning-Up for LC-Confirmation"

_foods, 2022, doi:10.3390/foods11081155_

Round 1
Reviewer 1 Report
See the attached revised pdf file

Author Response
Dear reviewing expert:
We would like to express our appreciation for your invaluable advice to our manuscript. We have revised the manuscript carefully according to the comments of the expert in the original text. And the details is report in the response letter.
Thanks with best wishes!
Shaoguan University
Jiebiao Guo 2022-4-2

Reviewer 2 Report
Authors have developed the Nano- affinity Cleaning-up method for the ochratoxin A detection which can be used as an alternative to the HPLC. The study is interesting and will catch the reader’s interest; however, I have several concerns that should be considered:
- There are too many abbreviations that make reading hard, therefore choose the abbreviation wisely. Moreover, in the abstract section abbreviations should be avoided for easy understanding.
- Name the full name of abbreviation while appearing first in the text e.g. LOD and LOQ.
- Introduction in lengthy containing many tables. I suggest pinpointing focus on the raised problem in the introduction section and shifting the tables and their content in the results and discussion part.
- Address the typo and grammatical mistake through the MS e.g 0.14 rather than 14 (line no. 164). Where is section 2.5 as mentioned in line 188 (material and methods)
- Mention the extraction buffer composition (line 186, 188).
- It would be better if the author can perform cross-reactivity against different mycotoxins to examine the sensitivity and practical utility of the assay.
- In figure 2: how by using TEM author come to know about AMA and AFR, is it possible??? If yes please explain or otherwise please correct this in figure 2 b. Further, correct the figure legend and the figure arrangement of Figures 2b and 2c. Similarly, in line 275 add figure no. appropriately.
- The values determined for zeta potential (+0.11 mV to -0.45 mV) (line 275) do not match with the values as depicted in the figure, is it the typo mistake (check and correct the values).
- In figure 3 b, how authors are so sure that each IFR is bound to IMA. What is the difference between figure 3b and 3c?
- Either mere discussion with results section or remove the subheading from the discussion section.
- Conclusion needs to write more logically emphasizing the current findings
Author Response
Dear reviewing expert:
We would like to express our appreciation for your invaluable advice to our manuscript. We have revised the manuscript carefully according to the advice point by point. The detail revision is reported in the response letter.
Thanks again with best wishes!
Best regards!
Shaoguan Unicersity
Jiebiao Guo 2022-4-2

Reviewer 3 Report
The comments are enclosed in the document

Author Response
Dear reviewing expert:
We would like to express our appreciation for your kind advice to our manuscript. We have revised the manuscript carefully according to the advice point by point. The detail revision is reported in the response letter.
Thanks again with best wishes!
Shaoguan Univisity
Jiebiao Guo
2022-4-2

Round 2
Reviewer 1 Report
After check of all the responses of the authors, I can support the acceptance of the manuscript.
Reviewer 2 Report
The manuscript has been revised. I give my consent to the article.